# Effects of Corn Supplementation into PKC-Urea Treated Rice Straw Basal Diet on Hematological, Biochemical Indices and Serum Mineral Level in Lambs

**DOI:** 10.3390/ani9100781

**Published:** 2019-10-11

**Authors:** Osama Anwer Saeed, Awis Qurni Sazili, Henny Akit, Abdul Razak Alimon, Anjas Asmara Samsudin

**Affiliations:** 1Department of Animal Science, Faculty of Agriculture, Universiti Putra Malaysia, Serdang, Selangor 43400, Malaysia; osama_anwr85@yahoo.com (O.A.S.); awis@upm.edu.my (A.Q.S.); henny@upm.edu.my (H.A.); 2Department of Animal Production, Faculty of Agriculture, University of Anbar, Anbar P.O Box 55431, Iraq; 3Faculty of Animal Science, Universitas Gadjah Mada, Yogjakarta 55281, Indonesia; razalimon@yahoo.co.uk

**Keywords:** hematology, biochemistry, corn, Dorper lambs, palm kernel cake

## Abstract

**Simple Summary:**

In Malaysia, livestock farms have used palm kernel cake as the main source of feed for livestock. The low cost and good content of crude protein in palm kernel cake makes it an appropriate food for sheep, but feeding huge amounts of palm kernel cakes (PKC) for a long period may expose them to copper toxicity. This study suggests supplementing a certain amount of corn as a source of energy into PKC basal diets estranged of toxicity to maintain the health status of lambs compared with other treatments, in terms of the hematological and biochemical parameters of lambs.

**Abstract:**

Twenty-seven Dorper lambs were used to determine the effect of supplementing corn as a source of energy into the palm kernel cake (PKC) urea-treated rice straw basal diet on the blood metabolic profile and metals in lambs. The lambs were randomly allotted to three experimental treatments according to their initial body weight for a 120 day trial. Dietary treatments were: T1 (control diet) = 75.3% of PKC + 0% corn, T2 = 70.3% of PKC + 5% corn, and T3 = 65.3% of PKC + 10% corn. The results of this study indicated that copper (Cu), selenium (Se), zinc (Zn), and iron (Fe) concentration intake, retention, and its absorption from the gut and apparent mineral digestibility were highly significant for the levels of corn supplementation. The biochemical and hematological parameters remained within normal levels with the treatments, but the white blood cell, eosinophil count, cholesterol, and low-density lipoprotein cholesterol (LDL) were significantly higher in T3. Treatment 3 significantly increased the concentration of Se and Fe, while Zn was reduced in the blood serum of lambs on day 120. The result shows that the inclusion of corn has no effect on the hematological and biochemical parameters of lambs after incorporating corn into the PKC-based diet at 5% and 10%.

## 1. Introduction

The high cost of grains for small ruminants due to competition with humans has led to the search for non-conventional sources. Several by-products of oil extraction can be used in animal feed without undesirable effects on the general performance of the animal [1]. Palm kernel cakes (PKC) serve, due to its abundance and availability in Malaysia [2,3], as a cost-effective source of protein in formulating rations for ruminants and non-ruminants alike. However, it has been reported that the inclusion of PKC in sheep rations greater than 80% for a long feeding time is associated with copper (Cu) toxicity, which has a detrimental effect on the liver, especially in crossbreed sheep. The inclusion of 15–35% level of concentrated PKC is optimal for goat feed. Furthermore, feeding PKC to lambs significantly increases the serum, liver, and meat iron (Fe) concentration (215–251, 49–66, and 14–18 ppm, respectively). It was also found to significantly contribute to the rise in the serum and liver zinc (Zn) concentration (101–130, 35–41, and 26–27 ppm, respectively) [4]. For the determination of metabolic blood profiles, including serum mineral and biochemical parameters, it is necessary to study ruminant metabolism disorders because blood serves as an indicator of the health status of animals and it is normal to assess the cause of an abnormality or malfunction of an animal by examining its blood. Correspondingly, the mineral content and biochemical indicators in the blood of sheep are widely used [5]. Blood contains diagnostically relevant parameters that act as a pathological reflector of the status of animals exposed to toxicants [6]. In reports, there seems to be a paucity of supplementing corn in PKC diets on the status of micro-minerals in the blood of sheep affected by Cu toxicity, which leads to the deterioration of the blood components of sheep. Moreover, based on the published literature, the proportion of PKC that should be utilized is controversial, mainly because of the mineral and fiber content that alters the replacement levels of PKC in the diets of animals. The combination of corn as a source of energy with different levels of PKC is hypothesized to alleviate the negative effect of feeding PKC to sheep, due to its high copper content. Thus, the present study aims to determine the influence of including corn as a source of energy into the PKC-based diet on the blood metabolic profile and the trace element status in the blood of lambs.

## 2. Materials and Methods 

### 2.1. Experimental Animals

Animal experimental procedures were conducted according to the Institutional Committee on Animal Use Ethics (Approval No. R064/2016). A total of 27 lambs (Dorper sheep) with an average live weight of 15 ± 0.59 kg were used for this study. The study was conducted at the small ruminant section of the livestock farm in Universiti Putra Malaysia. The lambs were housed in separate pens (1.20 m × 0.80 m × 0.70 m) equipped with feeders and drinkers. Water was provided *ad libitum* to the lambs. Ethics approval: It is confirmed that Institutional Animal Care and Use Committee (IACUC), Universiti Putra Malaysia approved this study.

### 2.2. Grouping of Animals

Animals were randomly assigned to three treatments and were fed three levels of corn at 0%, 5%, and 10%. Treatment 1 contained 75.3% PKC and 0% corn as the control diet, while Treatment 2 contained 70.3% PKC and 5% corn, and Treatment 3 included 65.3% PKC and 10% corn. Diets were formulated based on the recommendation of Reference [7] to meet the nutritional requirements of growing lambs. The theoretical assumptions for this formulation was based on meeting the body requirements of sheep in terms of protein (15.1%), energy (metabolizable energy 2.8 Mcal/kg), and the giving of a high level of PKC to the lambs and watching the physiological changes to the body. These diets were approximately isonitrogenous. Mineral premix was excluded in the diets in order to reduce the amount of Cu, as PKC has a high content of minerals.

### 2.3. Collection of Blood Samples

Blood (10 mL each) collected from the jugular vein was added into both serums of Vacutainer tubes) and ethylene diamine tetra-acetic acid (EDTA) (BD Franklin Lakes, NJ, USA) before feeding the lambs between 09:00 am and 11:00 am at the beginning of the feeding trial (d 0) and after 40, 80, and 120 days of the experiment. The EDTA tubes (2 mL each) were inverted several times to ensure an adequate mixing of the blood with the anticoagulant. After that, the serum was separated by centrifugation (10 min) at 2000 g at 4 °C. The serum was stored at −20 °C for biochemical measurements. Blood samples were processed immediately for hemogram analysis to check the general health of the lambs.

### 2.4. Hematological and Biochemical Evaluations

Hematological parameters were evaluated using a hematology analyzer (CELL-DYN 3700 Abbott, Illinois, USA). The biochemical indices such as total glucose, triglycerides, cholesterol, high-density lipoprotein cholesterol (HDL), low-density lipoprotein cholesterol (LDL), and very low-density lipoprotein cholesterol (VLDL) concentrations were measured by the colorimetric method [8]. Aspartate aminotransferase (AST) and alanine aminotransferase (ALT) activities were measured in the Roche Hitachi Diagnostic Modular Analyzer P-800 (Diamond Diagnostic, Massachusetts, USA) by using commercially available diagnostic kits supplied from Roche Diagnostics GmbH (D-68298, Mannheim, Germany).

### 2.5. Mineral Assay

Samples were digested according to Kolmer et al. [9] method. A total of 0.5 mL of the serum sample was poured into a digestion flask and 5 mL of nitric acid (HNO_3_) was added to it. The flask was then placed on a hot plate for 10–15 min until the fumes evaporated and 1–2 mL of liquid was left in the flask. After allowing the flask to cool down, 2.5 mL of perchloric acid (HClO_4_) was added to it. The solution in the flask was heated vigorously until the volume again reduced to 1–2 mL. The contents were then filtered and diluted to make up a total volume of 25 mL by adding deionized water and was stored in plastic bottles for analysis. Each digested and diluted sample was then used for the determination of Cu, Fe, Zn, and selenium (Se). All the samples (feed, refusals, feces, and urine) were wetted and ashed in 10 mL of HNO_3_ and 5 mL of HClO_4_, and analyzed for mineral elements using the same procedure with an increased amount of (HNO_3_) and (HClO_4_). The digested samples (Cu, Se, Zn, and Fe) were diluted with deionized triple glass distilled water. The Cu, Zn, and Fe concentrations were determined using an inductively coupled plasma atomic emission spectrometer (ICP-OES) (model Optima 8300, Perkin Elmer Inc., Massachusetts, USA) to determine the mineral content. Due to the high selectivity and sensitivity necessary to determine the Se concentration, the Se was measured using an ICP mass spectrometer ELAN Dynamic Reaction Cells (DRC)-e axial field technology (Perkin Elmer SCIEX, Waltham, United Kingdom) using the previous standards of Se.

The calculation of intake, urinary and fecal excretion, apparent absorption, retention, and apparent digestibility were done according to Wang and Fisher [10].

### 2.6. Statistical Analysis

Analyses were made in triplicate and the data obtained were expressed as means, followed by Analyses of Variance (ANOVA) for a completely randomized design using the Statistical Analysis Software (version 9.04). The analysis of variance and Duncan’s multiple range tests were used to determine significant differences (*p* < 0.01 and 0.05) among treatments. The variances among treatments were valued by using the following model:Y*ij* = μ + α*i* + *eij*
in which Y*ij* = dependent variable; μ = overall mean; α*i* = the fixed effect for supplementation of the level of corn (5 and 10%); and *e*ij = experimental error that was assumed to be normally and independently distributed (NID) with (0, *σ2* e).

## 3. Results

### 3.1. Minerals Balance

The treatment diets content of CP was 15.42%, 14.88%, and 14.09%, respectively, on a dry matter (DM) basis. The chemical composition and mineral content of the treatment diets are shown in Table 1.

Copper concentration intake and retention, and its absorption from the gut and apparent mineral digestibility, were significantly affected by the supplementation of corn into diets compared with control diets, while there was no effect between T2 and T3 (*p* < 0.01) (Table 2). A significant increase in the concentration of Cu was recorded in lambs fed T2 and T3 in the parameters of the study, while the excretion of Cu in fecal matter decline was significant (*p* < 0.01) with the inclusion of corn in the dietary treatment. The total excretion of Cu had no significant difference among the treatment groups, but the amount of Cu retained increased gradually (*p* < 0.01) in response to Cu levels in both corn diets.

However, the Se balance during the 120 days of the study showed significant differences among treatments, as seen in Table 3. Significant difference in Se intake was recorded at its highest in T3 (*p* < 0.001), while T2 had the lowest Se intake. The execration of Se in fecal matter and urine was not similar among all treatments, although it tended to be high in T3 lambs (*p* < 0.001). A significant increase in the levels of Se shows apparent absorption, retention, mineral digestibility, and mineral balance (*p* < 0.01) in lambs fed on T3 when compared with T1.

The Zn balance of lambs fed with different levels of corn diets is presented in Table 4. The lambs fed with corn treatment diets (T2 and T3) had lower (*p* < 0.001) Zn intake and Zn excretion in feces than those fed on T1. However, excretion of Zn via urine was not significantly different (*p* > 0.05) between treatment groups. The inclusion of 5% and 10% corn into the treatment diets were significantly lower (*p* < 0.001) in the apparent absorption and retention of Zn in lambs fed with these diets. The apparent mineral digestibility and mineral balance recorded a high percentage (*p* < 0.001) in T2 and T3 until the T1.

The concentration of Fe balance among treatments is shown in Table 5. The Fe intake and fecal excretion were highly significant (*p* < 0.001) in T1 compared with other diets. The level of Fe excreted in the urine was also significant (*p* < 0.001) and increased with the inclusion of corn in T2 and T3. However, the apparent digestibility and retention of Fe were significantly different among treatments (*p* < 0.001) with T2 and T3, which had a lower quantity of Fe compared to T1. In the apparent mineral digestibility and mineral balance of Fe, there was a highly significant difference (*p* < 0.01) among treatments, with the highest and lowest observed in the T3 and T2 treatments, respectively.

### 3.2. Hematological and Biochemical Parameters

The data on hematological parameters from blood collected in this experiment are presented in Table 6. The level of white blood cells (WBC) was higher (*p* < 0.05) in T3, while T2 did not differ when compared to T1. No significant variation (*p* > 0.05) in red blood cells (RBC), Hemoglobin (Hb), packed cell volume (PCV), mean corpuscular volume (MCV), mean corpuscular hemoglobin concentration (MCHC), B neutrophils (B Neut), S neutrophils (S Neut), and basophils (Baso) thrombocytes (Thrombo) were detected among the treatment groups. On the contrary, lymphocytes (Lymp), monocytes (Mono), and eosinophils (Eosin) concentrations in lambs fed with the T3 diet was higher when compared with lambs fed with the T1 diet.

The serum biochemistry of the Dorper lambs fed with the treatment diets are shown in Table 7. The values obtained were not significantly different (*p* > 0.05) among the three treatment diets fed to the lambs in some of the blood parameters measured. The serum’s total cholesterol level showed significant differences (*p* < 0.05) among treatments, which was higher in T3, but did not differ in T2 compared to T1. The highest LDL-cholesterol was recorded in T3, whereas T2 and T1 had the smallest (*p* < 0.05).

### 3.3. Minerals of Serum

The concentration of serum minerals measured at 0, 40, 80, and 120 days are shown in Figure 1. Dietary treatments at day 0 did not affect (*p* > 0.05) the level of retention of Cu, Se, Zn, and Fe in lambs. There were no effects (*p* > 0.05) of treatments on the concentration of Cu, Se, and Zn in blood serum, but for Fe retention, higher retention occurred in lambs fed on T1 at day 40. The recorded values were consistent at day 80 of the study and no differences were observed between lambs fed with corn at that time. Exhibiting the same trend, on the 120^th^ day, the data revealed that serum Cu was not significantly affected (*p* > 0.05) by the treatment. However, the Se level in the blood serum was higher (*p* < 0.001) in lambs fed with T3 than in those fed with only PKC urea-treated rice straw (T1). The serum concentration of Zn was lower (*p* < 0.05) for T2 and T3 than for the standard diet, while the differences between the two diets tended to small. Serum retention of Fe in lambs were different (*p* < 0.05), and higher Fe serum levels were noticed in lambs fed on the T3 diet, followed the by T2 and T1.

## 4. Discussion

### 4.1. Minerals Balance

Based on the values obtained in this study, dietary treatments had significant effects on Cu, Se, Zn, and Fe balance among treatment groups. The feeding of 5% corn along with 70.3% PKC or 10% corn with 65.3% PKC resulted in higher Cu intake, apparent absorption, retention, apparent mineral digestibility, and mineral balance compared to no corn diets (T1). The increased consumption of the diets is due to the rise in Cu intake and this finding is in line with Oladokun et al. [11]. The increase in dietary Cu caused the groups difference to diminish over time, indicating that any group distinction in the metabolic process is probably saturated with high Cu intakes [12]. With regards to the Cu balance, the amount of its excretion in fecal matter was greater than in urine due to the primary route of trace mineral excretion being through the feces [13]. Generally, the quantity of Cu excreted in the fecal matter of lambs fed on T2 and T3 could transform Cu into a complex compound of protozoa, resulting in it being excreted less by lambs than lambs not fed the corn diet. Ivan et al. [14] clarified that the increase in Cu bioavailability due to the elimination of ciliate protozoa from the rumen may amount to between 15% and 50%. However, Suttle [15] reported that sulfur could be converted to sulfide by ruminal bacteria and could have a similar adverse effects on the bioavailability of Cu.

In our experiment, the intake, absorption, retention, apparent mineral digestibility, and mineral balance of Se was high in T3, which is a marked diet effect, since the high content of PKC improves the availability of Se [16,17]. However, the low absorption of Se in lambs is the result of a reduction of dietary Se to insoluble forms such as elemental Se or hydrogen selenide in the rumen environment [17]. The differences in the absorption among the groups depends on the chemical form of Se and the composition of the diet [16]. The amount of Se did not cause toxicity problems, and the maximum tolerable amount is 5 mg Se/kg DM [18]. Most of the inorganic Se is not used immediately in the liver for selenoprotein synthesis and is quickly excreted via the urine [17]. This explains the increase in the level of Se secretion in the urine of lambs in T1.

In the present experiment, Zn consumption, feces Zn excretion, apparent absorption, and retention in lambs fed with the corn supplemented diet were lower than that of the control group, but the apparent mineral digestibility of Zn (58.31%) and Zn balance (60.69%) was higher due to the increase in fecal Zn excretion and the overall mean of Zn excretion. However, the high intake of divalent cations such as Cu, Fe, and Ca reduced Zn absorption [18]. The amount of Zn excreted via urine excretion was similar between lambs fed with corn and those on control diets. The results also show that the urinary excretion of Zn by lambs were generally higher than 1 mg/d with no effect on secretion due to its concentration in these diets [18]. It was previously reported that dietary Zn reduced the accumulation of hepatic Cu and promotes the formation of relatively non-toxic forms of Cu in the liver, such as metallothionein, which is involved in the storage and detoxification of Cu and other heavy metals [19,20]. In the present experiment, the supplemented corn diet resulted in relatively low dietary Zn concentrations. Additionally, it was much less than 1 g/kg DM, leading to the lower feed intake and growth of lambs [21,22].

The intake of Fe by lambs fed with the treated diet decreased dramatically. McDowell [23] reported that the massive amount of Zn and Fe intake could affect Cu utilization. High dietary Zn and Fe can reduce Cu absorption in cattle and sheep [24,25]. The primary routes of Fe excretion are via feces and urine [23]. In this study, corn supplementation had an effect on the dietary level of Fe, which altered its absorption amount, resulting in a decrease in its absorbable intake. In any event, the increase in apparent Fe uptake tends to be associated with high Fe retention. It is likely that there exists a mineral binding protein in the PKC diet treated with corn, which is responsible for the decreased absorption of Fe via the process of maintaining its ions in a soluble form during its transfer from the lumen of the intestine into the cell of the intestinal wall. The concentration of Fe in PKC is known to range between 800 and 6000 mg/kg DM and was 640–1800 mg/kg DM in this study. The study by Al-Kirshi et al. [26] indicated the probability of using a Fe diet to reduce the bioavailability of dietary Cu because of the high PKC content.

### 4.2. Hematological and Biochemical Parameters

The average concentrations of hemogram and biochemical indicators in the blood of lambs remained normal at the beginning of the experiment, which is based on average values for RBC concentrations in sheep (8–13 (× 10^6^) RBC/mL) [27]. The MCV values of the treatments were within the normal range of 23–48 fl [28]. The variations observed agree with Daramola et al. [29] who reported that age was a practical factor that has a significant effect on the Hb and RBC of goats, suggesting that the oxygen-carrying capacity of the blood was high in adult goats.

The unchanged PCV, MCV, and MCHC in lambs fed diets with or without corn is supported by the findings of Galıp [30]. The variations in the blood profile occurred as time elapsed, which is due to changes in physiology resulting in the growth of ruminants [29,31] where it was found that the age of farm animals affects their hematological parameters.

The values of WBC, which can be used as an indicator of inflammation, were significantly different among groups. The mean concentration of WBC of the groups remained within the normal range (4000–12,000 × 10^9^/L) [27,28]. The absence of any effect on B Neuts and S Neuts indicated that the source of protein has no effect on the blood profile. The results of this study showed that using PKC as a protein source had no adverse effects on the blood profile and this finding is supported by Nelson and Watkins [32], who indicated that variations in protein sources might not influence the homeostatic mechanism. The lymphocytes, monocytes, and basophils were affected by the treatment, which increased in the group fed on T3. The reason for this is not clear and no corresponding information is available.

The serum ALT, ALP, and AST levels did not indicate significant differences between the treatment means. Total cholesterol levels were significantly different, and there was an increasing trend in T2 and T3, suggesting a dietary influence although the HDL and VLDL in the serum did not register any significant difference among the treatments, while LDL increased significantly in T3. This observation is explained by the fact that the cholesterol content in the serum has been used to assess the changes in lipid metabolism by feeding oil diets. The simultaneous increase in cholesterol and HDL levels in the serum of lambs with a corn-supplemented PKC diet in this study indicates that the inclusion was clearly reflected in the serum cholesterol level and that the increased cholesterol was mainly due to the increase in HDL level [31,33,34]. The serum level of triglycerides and glucose in lambs fed with corn diets recorded non-significant variations among treatments. In addition, observed blood glucose concentrations were similar to those overseved by Turner et al. [35], although variations in them could be affected by physiological status. In this study, the findings indicate that lambs fed on diets with corn in the PKC were in the normal energy status range. This could be a contributory factor for the lack of differences among treatments, and there were no negative effects on feed intake or the metabolism of the lambs. The serum metabolite values seen in this study were all within the normal range for sheep serum, as reported by Pampori [36]. However, in spite of the higher figures recorded in this study, the corn-fed PKC diets did not have adverse effects on the health conditions of the lambs.

### 4.3. Mineral Serum 

This research shows that the dietary inclusion with different corn levels affected the Cu, Se, Zn, and Fe content in the blood serum of lambs. It shows that the Cu concentration in the lambs’ serum was below critical levels, indicative of a sufficient dietary treatment. Therefore, the micro minerals’ mean value of serum Cu is similar to the findings of Hidiroglou et al. [37] and Karim and Verma [38]. The average Cu in the serum from experimental animals was near the reference values. The previous study showed no significant effect of feeding TMR with or without PKC on the Cu concentration in the serum [4].

However, wide variations have been observed in the Se concentration among lambs fed on T3 and this is generally considered an indicator of body health. This linear relationship between blood Se and Se intake suggests that the former may be an adequate marker for Se status over a range of intakes. Chalabis–Mazurek and Walkuska [39] noticed that Se supplementation distinctly depressed the Cu and Fe content in the serum and liver in lambs. Increased feed intake in T3, leading to high concentrations of Se in lambs, seems to be beneficial with respect to its function as an oxidant contained in glutathione peroxidase, an activity that is conditioned by the presence of Se.

Feeding PKC significantly increased Zn and Fe concentrations in the serum and this finding is in line with Abdelrahman et al. [4]. No difference was seen in lambs receiving the experimental diets (T2 and T3) at day 40 and 80, but for lambs that were fed on T1 for 120 days, serum Zn concentrations increased, while being depressed in the treatment diets. The decline in Zn levels in T2 and T3 could be due to improvements in the immune body of the lambs fed on T2 and T3. Kargin et al. [40] noted that Zn is an essential element required for the immunity function to play a role in responding to diseases. Active, synergistic, or antagonistic interrelationships between Fe and other minerals such as Cu and Zn have been documented [41,42,43,44]. Throughout the experiment, the content of the serum Fe fluctuated, showing an increasing tendency after 10 weeks of the supplementation treatment. However, statistically significant changes were noted in the lamb group that was given 10% corn supplementation. Accordingly, high intakes of Fe by lambs fed PKC could affect their hormonal activities, and consequently, affect their metabolism and health [4]. This finding could provide a good indicator of the body health of animals. It has been reported that a high level intake of Cu, Mg, and Zn contributed to reducing the absorption of Fe [45]. Further, trace minerals like Fe and Mn have also been shown to be Se antagonists [46].

The concentration of all trace minerals reported in this study falls within the normal range, except for Fe values, which were found to be above the normal range in lambs fed with PKC with or without corn. Variations in the mineral composition of PKC among different scientists might be due to the type of soil, water, sampling techniques, and assay procedures [47].

## 5. Conclusions

It is evident from the present study that the inclusion of corn into a PKC urea-treated rice straw-based diet does not have an effect on the hematological and biochemical parameters of lambs after incorporating corn as a source of energy into a PKC-based diet at 5% and 10%. It could be concluded that the non-significant changes detected in the serum Cu reflects the health status of lambs among groups that related to no difference in liver enzymes. Therefore, these diets could minimize the feed costs and improve the profitability of farming; thus, small ruminant nutrition researchers may benefit by conducting further researches.

## Figures and Tables

**Figure 1 animals-09-00781-f001:**
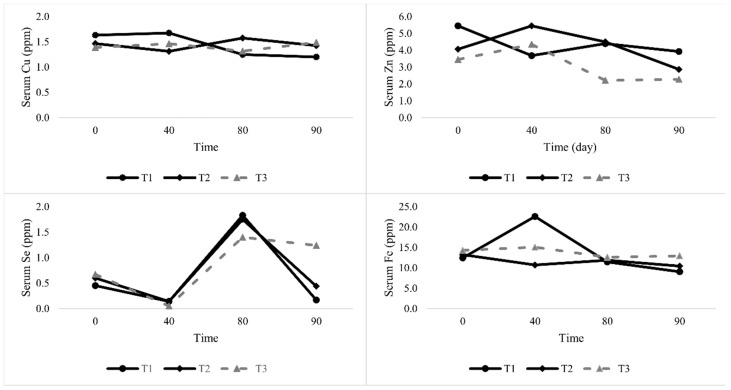
Effect of experimental diets on some mineral levels in the serum (ppm) of lambs at different times. T1: (75.3% PKC + 0% corn), T2: (70.3% PKC + 5% corn), and T3: (65.3% PKC + 10% corn). PKC = palm kernel cake.

**Table 1 animals-09-00781-t001:** Chemical composition and approximate analysis minerals (mg/kg dry matter (DM)) of experimental diets.

Parameters	Treatments		
	T1	T2	T3
DM	91.78	91.66	91.55
Ash	13.80	12.72	12.74
OM	86.19	87.27	87.26
CP	15.42	14.88	14.09
EE	5.3	5.1	4.33
CF	26.6	24.50	20.83
NDF	62.36	60.06	55.66
ADF	45.60	40.96	37.30
ADL	6.56	6.10	5.43
ME MJ/Kg DM	7.36	8.23	8.92
Se	0.36	0.34	0.29
Cu	6.16	5.96	5.80
Zn	26.25	14.65	11.28
Fe	1869.7	761.3	643.4

T1: (75.3% PKC + 0% corn), T2: (70.3% PKC + 5% corn), and T3: (65.3% PKC + 10% corn). Vitamin premix; Vitamin A: 10,000,000 IU; Vitamin E: 70,000 IU; Vitamin D: 1,600,000 IU; DM: dry matter; OM: organic matter; CP: crude protein; EE: ether extract; CF: crude fiber; NDF: neutral detergent fiber; ADF: acid detergent fiber; ADL: acid detergent lignin; and ME: metabolizable energy.

**Table 2 animals-09-00781-t002:** Copper balance by Dorper lambs fed with different levels of corn.

Item	Treatments				
	T1	T2	T3	SEM	*p*-Value
Intake (mg/d)	5.73 ^B^	7.41 ^A^	7.39 ^A^	0.08	**
Excretion (mg/d)					
Fecal	4.19 ^A^	3.64 ^B^	3.60 ^B^	0.09	**
Urinary	0.64 ^AB^	0.36 ^B^	0.90 ^A^	0.09	**
Total	3.76	3.22	3.25	0.15	NS
Apparent absorption					
(mg/d)	1.84 ^B^	3.76 ^A^	3.78 ^A^	0.08	**
of intake %	28.27 ^B^	48.38 ^A^	48.28 ^A^	0.72	**
Retention					
(mg/d)	2.06 ^B^	4.19 ^A^	4.14 ^A^	0.08	**
of intake %	34.40 ^B^	54.45 ^A^	53.34 ^A^	0.68	**
Apparent mineral digestibility (%)	28.67 ^B^	48.38 ^A^	48.28 ^A^	0.71	**
Mineral balance (%)	34.08 ^B^	54.45 ^A^	53.34 ^A^	0.70	**

T1: (75.3% PKC + 0% corn), T2: (70.3% PKC + 5% corn), and T3: (65.3% PKC + 10% corn). Different capital letters indicate statistically significant differences among the treatment groups (*p* < 0.01). NS: not significant statistically (*p* > 0.05), ** *p* < 0.01.

**Table 3 animals-09-00781-t003:** Selenium balance by Dorper lambs fed with different levels of corn.

Item	Treatments				
	T1	T2	T3	SEM	*p*-Value
Intake (mg/d)	0.33 ^B^	0.31 ^C^	0.37 ^A^	0.003	***
Excretion (mg/d)					
Fecal	0.15a ^B^	0.16 ^A^	0.13 ^B^	0.003	***
Urinary	0.522 ^A^	0.035 ^B^	0.012 ^B^	0.105	***
Total	0.18 ^A^	0.14 ^B^	0.12 ^B^	0.009	**
Apparent absorption					
mg / d	0.18 ^B^	0.15 ^C^	0.24 ^A^	0.004	***
of intake %	52.28 ^B^	46.97 ^C^	62.58 ^A^	0.606	***
Retention					
mg/d	0.15 ^B^	0.16 ^B^	0.25 ^A^	0.004	***
of intake %	43.24 ^C^	50.77 ^B^	65.66 ^A^	0.672	***
Apparent mineral digestibility (%)	52.88 ^B^	46.97 ^C^	62.58 ^A^	0.591	**
Mineral balance (%)	44.04 ^C^	50.77 ^B^	65.66 ^A^	0.700	**

T1: (75.3% PKC + 0% corn), T2: (70.3% PKC + 5% corn), and T3: (65.3% PKC + 10% corn). Different capital letters indicate statistically significant differences among the treatment groups (*p* < 0.01 and *p* < 0.001). NS: not significant statistically (*p* > 0.05) ** *p* < 0.01 and *** *p* < 0.001.

**Table 4 animals-09-00781-t004:** Zinc balance by Dorper lambs fed with different levels of corn.

Item	Treatments				
	T1	T2	T3	SEM	*p*-Value
Intake (mg/d)	24.41 ^A^	13.63 ^B^	14.39 ^B^	0.30	***
Excretion (mg/d)					
Fecal	13.67 ^A^	7.23 ^B^	5.65^C^	0.50	***
Urinary	2.10	1.81	2.23	0.17	NS
Total	12.16 ^A^	6.52 ^B^	5.34 ^B^	0.51	***
Apparent absorption					
mg/d	11.52 ^A^	6.39^C^	8.73 ^B^	0.20	***
of intake %	43.55 ^B^	44.35 ^B^	58.31 ^A^	0.62	***
Retention					
mg/d	12.42 ^A^	7.11^C^	9.04 ^B^	0.22	***
of intake %	48.00 ^B^	49.82 ^B^	60.60 ^A^	0.58	***
Apparent mineral digestibility (%)	43.24 ^B^	44.83 ^B^	58.31 ^A^	0.62	***
Mineral balance (%)	49.02 ^B^	49.82 ^B^	60.60 ^A^	0.56	***

T1: (75.3% PKC + 0% corn), T2: (70.3% PKC + 5% corn), and T3: (65.3% PKC + 10% corn). Different capital letters indicate statistically significant differences among the treatment groups (*p* < 0.001). NS: not significant statistically (*p* > 0.05) and *** *p* < 0.001.

**Table 5 animals-09-00781-t005:** Iron balance by Doper lambs fed with different levels of corn.

Item	Treatments				
	T1	T2	T3	SEM	*p*-value
Intake (mg/d)	1739.10 ^A^	708.57 ^C^	820.54 ^B^	25.13	***
Excretion (mg/d)					
Fecal	889.46 ^A^	461.16 ^B^	336.95 ^C^	33.95	***
Urinary	7.01 ^B^	15.45 ^B^	170.14 ^A^	29.78	***
Total	774.36 ^A^	420.64 ^B^	321.78 ^B^	35.99	***
Apparent absorption					
mg/d	855.95 ^A^	271.72 ^C^	476.11 ^B^	17.32	***
of intake %	47.96 ^B^	35.36 ^C^	56.19 ^A^	0.66	***
Retention					
mg/d	977.03 ^A^	294.29 ^C^	491.64 ^B^	18.48	***
of intake %	53.54 ^B^	39.05 ^C^	58.17 ^A^	0.65	***
Apparent mineral digestibility (%)	48.18 ^B^	33.97 ^C^	56.45 ^A^	0.69	**
Mineral balance (%)	54.45 ^B^	38.77 ^C^	58.17 ^A^	0.65	**

T1: (75.3% PKC + 0% corn), T2: (70.3% PKC + 5% corn), and T3: (65.3% PKC + 10% corn). Different capital letters indicate statistically significant differences among the treatment groups (*p* < 0.01 and *p* < 0.001). NS: not significant statistically (*p* > 0.05) ** *p* < 0.01 and *** *p* < 0.001.

**Table 6 animals-09-00781-t006:** The effect of PKC-urea treated rice straw supplemented with different levels of corn on the hematological blood parameters in lambs.

Parameters	Treatments				
	T1	T2	T3	SEM	*p*-value
RBC ×10^12^/L	10.51	9.98	10.15	0.19	NS
Hb g/L	104.65	103.25	103.88	1.45	NS
PCV L/L	0.293	0.292	0.286	0.004	NS
MCV fL	28.12	29.40	28.53	0.33	NS
MCHC g/L	357.20	353.51	363.04	1.68	NS
WBC × 10^9^/L	7.48 ^b^	7.82 ^ab^	8.81 ^a^	0.22	*
B Neut ×10^9^/L	0.080	0.102	0.095	0.004	NS
S Neut ×10^9^/L	3.40	3.72	3.68	0.13	NS
Lymp ×10^9^/L	3.399 ^b^	3.398 ^b^	4.101 ^a^	0.12	*
Mono ×10^9^/L	0.387 ^b^	0.401 ^b^	0.504 ^a^	0.01	*
Eosin ×10^9^/L	0.147 ^B^	0.148 ^B^	0.298 ^A^	0.01	***
Baso ×10^9^/L	0.091	0.101	0.141	0.009	NS
Thrombo	680.07	725.12	831.29	34.85	NS

T1: (75.3% PKC + 0% corn), T2: (70.3% PKC + 5% corn), and T3: (65.3% PKC + 10% corn). Different capital letters indicate statistically significant differences among the treatment groups (*p* < 0.001). Different lowercase letters indicate statistically significant differences among the treatment groups (*p* < 0.05). NS: not significant statistically (*p* > 0.05) * *p* < 0.05 and *** *p* < 0.001. PKC = palm kernel cake; RBC = red blood cell; PCV = packed cell volume; MCV = mean corpuscular volume; MCHC = mean corpuscular hemoglobin concentration; and WBC = white blood cell.

**Table 7 animals-09-00781-t007:** Effect of PKC-urea treated rice straw supplemented with different levels of corn on the biochemical blood parameters in lambs.

Parameters	Treatments				
	T1	T2	T3	SEM	*p*-Value
ALT U/L	15.47	12.81	11.81	0.79	NS
ALP U/L	151.80	163.45	152.80	8.01	NS
AST U/L	115.30	103.67	107.91	2.55	NS
Cholesterol, mmol/L	2.52 ^b^	2.79 ^ab^	3.13 ^a^	0.10	*
Glucose, mmol/L	3.22	3.40	3.47	0.10	NS
Triglyceride, mmol/L	0.61	0.58	0.60	0.03	NS
LDL-cholesterol, mmol/L	0.575 ^b^	0.571 ^b^	0.726 ^a^	0.03	*
HDL-cholesterol, mmol/L	1.82	2.11	2.28	0.09	NS
VLDL-cholesterol, mmol/L	0.22	0.21	0.26	0.01	NS

T1: (75.3% PKC + 0% corn), T2: (70.3% PKC + 5% corn), and T3: (65.3% PKC + 10% corn). Different lowercase letters indicate statistically significant differences among the treatment groups (*p* < 0.05). NS: not significant statistically (*p* > 0.05) and * *p* < 0.05. ALT = alanine transaminase; ALP = alkaline phosphatase; AST = aspartate aminotransferase; LDL = low density lipoprotein; HDL = high density lipoprotein; and VLDL = very low density lipoprotein.

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
