# Peer review of "Effects of Corn Supplementation into PKC-Urea Treated Rice Straw Basal Diet on Hematological, Biochemical Indices and Serum Mineral Level in Lambs"

_animals, 2019, doi:10.3390/ani9100781_

Round 1

Reviewer 1 Report

Comments are reported in the boxes near text highlighted in yellow

Author Response

Dear Editor,

We would like to thank the journal and the reviewers for the valuable and useful comments for our manuscript. 

We have made the following corrections/modifications and additions to our manuscript. Points addressing the specific comments raised by the reviewers, and the detailed responses are listed in the following table:

Note: Corrections for Reviewer 1 has highlighted with yellow color

No.

Comments

Corrections made

Page / Line No.

Reviewer 1

1

Rewrite unclear

The correction has been made as you suggested

Materials and Methods, Page 4/ 114-121

2

Report in apposite table the chemical composition of 3 diets

The chemical composition of three diets has been added, the table shows the level of crude protein is almost similar among diets.

Results, Page 3-4/ 124-130

3

This sentence is uncorrected! In fact only T1 intake is significantly lower than T2 and T3, but there are not significant differences between T2 and T3 diets. Rewrite this sentence.

The correction has been made as you suggested

Results, Page 4 / 132 -134

4

This sentence is uncorrected, T1 vs T2 Urinary excretion don't differ statistically.

Correction has been made accordingly

Results, Page 4 / 137 -139

5

Why authors report ** (P<0.01) if the differences between means are significant at P<0.05 (lowercase letters)?

The correction has been made as you suggested throughout the manuscript

6

Rewrite

Correction has been made

Results, Page 4 /144 – 145

7

In the international literature the lowercase letters indicate differences at P<0.05; different capital letters indicate differences at P<0.01. Please correct the tables.

The correction has been made as you suggested throughout the manuscript

8

This sentence is in contrast with data reported in table 2

Correction has been made

Discussion, Page 8 / 224-225

9

This sentence is uncorrected and unexplainable. T2 and T3 diets contains less Se than T1 (table 1). The NSC are not reported in any table.

The sentence has corrected, the table we reported in this paragraph is Table 3

Discussion, Page 8 / 236-237

10

This sentence cannot be reported because authors don't report in this paper a rumen microbiological study

The sentence has omitted as you suggested

Discussion, Page 9 / 253

11

Unclear sentence

Correction has been made

Discussion, Page 9 / 270-272

12

report the name

Correction has done

Discussion, Page 10 / 321

13

These conclusions are in contrast with results found. Moreover, abstract report different conclusion respect this section.

The correction has been made

Conclusion, Page, 11 / 341-345

We have endeavored to the best of our ability to address the points raised by the referees. It is hoped that the quality of the manuscript has improved and can now be considered for publication. We would like to thank the editors and reviewers once again for going through our responses.

Sincerely,

Authors

Reviewer 2 Report

The manuscript animals-574427 aimed to determine the effect of supplementing corn as a source of energy into palm kernel cake (PKC) urea-treated rice straw basal diet on blood metabolic profile and metals in lambs. The study is technically sound. However, authors are advised to check the English style of the manuscript, as there are many sentences that are difficult to understand.

·         Because of the grammar problems, it is very difficult to understand the rationale of the study

·         It is not clear what is the main hypothesis of the study – it should not be just the lack of research on the specific topic

Line 29 please revise sentence

Lines 29-31 check English grammar

Lines 38-40 check English grammar

Lines 46 give ranges for those increases in serum, liver and meat

Line 62 delete about

Line 63 give housing details i.e., pen dimensions, did animals had access to water?? Please give details

Please provide details on how did you formulate the diets? Based on NRC?? Or other nutritional guideline?

What were the theoretical assumptions for that formulation?

Line 71 at what time did blood samples were taken? Did you store samples at -20 or -80C?

Lines 96-99 please provide more details about your measurements.

Table 1. is it approximate or proximate? In either case, you need to add dry matter, NDF, ADF, lignin, ash, protein and fat.

Statistical analysis: provide statistical design and analysis, explain fixed and random variables

Figure 1, resolution needs to be improved

The discussion has a reasonable structure however; English grammar style should be checked.

Conclusion is confusing, please condense sentences and also in one sentence explain why your results are of value for farmers and small ruminant nutrition researchers

Author Response

Dear Editor,

We would like to thank the journal and the reviewers for the valuable and useful comments for our manuscript. 

We have made the following corrections/modifications and additions to our manuscript. Points addressing the specific comments raised by the reviewers, and the detailed responses are listed in the following table:

Note: Corrections for Reviewer 2 has highlighted with green color

Reviewer 2

Corrections made

Page / Line No.

1

It is not clear what is the main hypothesis of the study

The hypothesis has been improved as suggested.

Introduction

Page. 2/58-61

2

Please revise sentence

Correction has made

 Abstract, Page 1 / 30-32

3

Check English grammar

The correction has been made as you suggested

Introduction

Page. 2/40-42

4

Give ranges for those increases in serum, liver and meat

Corrected

Introduction

Page. 2/47-50

5

Delete about

The correction has been made in the paper as suggested.

Materials and Methods Page 2/67

6

Give housing details i.e., pen dimensions, did animals had access to water?? Please give details

The correction has been made in the paper as suggested

Materials and Methods Page 2/69-71

7

Please provide details on how did you formulate the diets? Based on NRC?? Or other nutritional guideline?

Correction has made

Materials and Methods Page 2/75-76

8

What were the theoretical assumptions for that formulation?

The theoretical assumptions for this formulation was based on meeting the body requirements of sheep in term of protein and energy and the giving high level of PKC to the sheep and watching the physiological changes of the body.

9

What time did blood samples were taken? Did you store samples at -20 or -80C?

Correction has made

Materials and Methods Page 2/82-86

10

Please provide more details about your measurements.

The correction has been made

Materials and Methods, Page 3/ 105-111

11

Statistical analysis: provide statistical design and analysis, explain fixed and random variables

The correction has been made as you suggested

Materials and Methods, Page 3/ 114-122

12

Table 1. is it approximate or proximate? In either case, you need to add dry matter, NDF, ADF, lignin, ash, protein and fat.

Its approximate analysis, the chemical composition of diets have been added as you recommended

Results  Page 3-4/124-131

13

Figure 1, resolution needs to be improved

14

The discussion has a reasonable structure however; English grammar style should be checked.

15

Conclusion is confusing, please condense sentences and also in one sentence explain why your results are of value for farmers and small ruminant nutrition researchers

The correction has been made

Conclusion, Page, 11 / 344-346

We have endeavored to the best of our ability to address the points raised by the referees. It is hoped that the quality of the manuscript has improved and can now be considered for publication. We would like to thank the editors and reviewers once again for going through our responses.

Sincerely,

Authors

Round 2

Reviewer 1 Report

Paper has been corrected and improved, now it is published in Animals Journal.

A moderate English revision is suggested too.

Author Response

Thanks for your comments. It's very useful for us.

Reviewer 2 Report

Authors have improved corrections however; some details need to be addressed:

Hypothesis statement: means that authors should describe what where the expected results based on the previous information they found from the specific topic. The authors wrote that there were some discrepancies between previous reports but they have not stated what will be the expected results if xxx is fed to lambs on xxxx conditions.

Authors need to include in the text the theoretical assumptions for the feed formulation: “The theoretical assumptions for this formulation was based on meeting the body requirements of sheep in term of protein and energy and the giving high level of PKC to the sheep and watching the physiological changes of the body.” Specifically they need to explain ranges of energy and protein for growing lambs.

Figure 1 needs better resolution; the authors have not changed that.

Reference 7. They is written is very atypical, please check with the journal’s regulations.
